# In-Hospital Patient Education Markedly Reduces Alcohol Consumption after Alcohol-Induced Acute Pancreatitis

**DOI:** 10.3390/nu14102131

**Published:** 2022-05-20

**Authors:** Rita Nagy, Klementina Ocskay, Alex Váradi, Mária Papp, Zsuzsanna Vitális, Ferenc Izbéki, Eszter Boros, László Gajdán, Andrea Szentesi, Bálint Erőss, Péter Jenő Hegyi, Áron Vincze, Judit Bajor, Patricia Sarlos, Alexandra Mikó, Katalin Márta, Dániel Pécsi, Andrea Párniczky, Péter Hegyi

**Affiliations:** 1Centre for Translational Medicine, Semmelweis University, 1085 Budapest, Hungary; drnagyrita@yahoo.com (R.N.); ocskay.klementina@gmail.com (K.O.); szentesiai@gmail.com (A.S.); dr.eross.balint@gmail.com (B.E.); katalin.martak@gmail.com (K.M.); andrea.parniczky@gmail.com (A.P.); 2Institute for Translational Medicine, Szentágothai Research Centre, Medical School, University of Pécs, 7624 Pécs, Hungary; varadi.alex09@gmail.com (A.V.); boroseszter987@gmail.com (E.B.); drdunajskastreda@gmail.com (P.J.H.); miko.szandi@gmail.com (A.M.); 3Heim Pál National Pediatric Institute, 1089 Budapest, Hungary; 4Division of Gastroenterology, Department of Internal Medicine, University of Debrecen, 4032 Debrecen, Hungary; papp.maria@med.unideb.hu (M.P.); vitalis@med.unideb.hu (Z.V.); 5Department of Internal Medicine, Szent György University Teaching Hospital of County Fejér, 8000 Székesfehérvár, Hungary; izbeki@mail.fmkorhaz.hu (F.I.); lgajdan@yahoo.com (L.G.); 6Centre for Translational Medicine, Department of Medicine, University of Szeged, 6720 Szeged, Hungary; 7Division of Pancreatic Diseases, Heart and Vascular Center, Semmelweis University, 1082 Budapest, Hungary; 8Division of Gastroenterology, First Department of Medicine, Medical School, University of Pécs, 7624 Pécs, Hungary; vincze.aron@pte.hu (Á.V.); bajor.judit@pte.hu (J.B.); sarlos.patricia@pte.hu (P.S.); daniel.pecsi1991@gmail.com (D.P.); 9Department of Medical Genetics, Medical School, University of Pécs, 7623 Pécs, Hungary

**Keywords:** brief intervention, acute pancreatitis, recurrence, alcohol, gamma-glutamyl transferase

## Abstract

Although excessive alcohol consumption is by far the most frequent cause of recurrent acute pancreatitis (AP) cases, specific therapy is still not well established to prevent recurrence. Generally, psychological therapy (e.g., brief intervention (BI)) is the cornerstone of cessation programs; however, it is not yet widely used in everyday practice. We conducted a post-hoc analysis of a prospectively collected database. Patients suffering from alcohol-induced AP between 2016 and 2021 received 30 min BI by a physician. Patient-reported alcohol consumption, serum gamma-glutamyl-transferase (GGT) level, and mean corpuscular volume (MCV) of red blood cells were collected on admission and at the 1-month follow-up visit to monitor patients’ drinking habits. Ninety-nine patients with alcohol-induced AP were enrolled in the study (mean age: 50 ± 11, 89% male). A significant decrease was detected both in mean GGT value (294 ± 251 U/L vs. 103 ± 113 U/L, *p* < 0.001) and in MCV level (93.7 ± 5.3 U/L vs. 92.1 ± 5.1 U/L, *p* < 0.001) in patients with elevated on-admission GGT levels. Notably, 79% of the patients (78/99) reported alcohol abstinence at the 1-month control visit. Brief intervention is an effective tool to reduce alcohol consumption and to prevent recurrent AP. Longitudinal randomized clinical studies are needed to identify the adequate structure and frequency of BIs in alcohol-induced AP.

## 1. Introduction

Acute pancreatitis (AP) is one of the most common gastrointestinal-system-related reasons for hospital admission affecting 13–80/100,000 people worldwide, with the primary causes of gallstone and excessive alcohol consumption [1,2,3]. It is very important to note that the disease can recur in 20–30% of cases, which can lead to further organ damage even in end-stage diseases such as chronic pancreatitis or pancreatic cancer [4,5]. Therefore, specific therapy for pancreatitis of different etiologies is of utmost importance. A significant proportion of etiologies have specific therapies to avoid recurrence, such as cholecystectomy in biliary etiology or fibrate or statin therapy in hyperlipidemic AP and steroid therapy in the case of autoimmune AP [6,7,8]. Unfortunately, alcohol-induced pancreatitis stands out in this field, and alcohol-induced AP is by far the most common form of RAP [9,10]; therefore, research on specific therapies for decreasing the number of recurrent alcohol-induced AP is of crucial importance [11,12].

Alcohol misuse is a wide spectrum ranging from binge drinking to alcohol use disorder (AUD), which is determined based on the criteria of the Diagnostic and Statistical Manual of Mental Disorders, fifth edition (DSM-5). Heavy drinking (14 or 7 drinks per week for men and women, respectively) and consequent recurrent inflammation can lead to permanent damage to the pancreatic tissue and facilitate the development of chronic pancreatitis [13,14]. Furthermore, excessive alcohol intake itself can decrease life expectancy by 24–28 years compared to the general population, as shown in Nordic countries [15]. Results of Razvodovsky et al. suggested that 63% of all male pancreatitis deaths in Russia could be attributed to alcohol consumption [16]. Despite the fact that heavy drinking and alcohol use disorder (AUD) are continuously spreading worldwide, leading to increased health, economic, and social burdens, there is a lack of intention to encourage patients either to participate in cessation programs or to keep long-term abstinence [17].

Since excessive alcohol consumption is a global emerging healthcare issue, there are several therapeutic options available for the initiation of cessation of alcohol, including psychosocial and pharmacological interventions. Alcohol misuse definitely must be highlighted since alcohol is responsible for 1 out of 7 male and 1 out of 13 female deaths in the age range of 15–64 years in the European Union [18]. Generally, the psychological approach is the cornerstone of cessation programs often combined with pharmacological treatment to achieve the most favorable results [19]. Following motivation assessment, brief intervention is generally the first step provided to patients. Brief interventions (BIs), a type of psychological intervention that ranges from 5 to 30 min, aim to highlight the fact of risky alcohol consumption and its negative effects and emphasize patients’ responsibility to quit [20]. One of the commonly used tools in BI is the FRAMES model, which systematically summarizes the main six elements (Feedback, Responsibility, Advice, Menu for change, Empathy, and enhancing Self-efficacy). Brief interventions have already been proven beneficial in clinical practice by several studies [21,22,23]. A meta-analysis of 52 trials by Platt et al. showed that BIs significantly reduced the amount of alcohol intake; later, this finding was confirmed in a study by the Cochrane Collaboration [22,23]. A randomized clinical trial by Nordback et al. published in 2009 found that repeated BIs can lower the recurrence rate of alcohol-induced AP [21].

To confirm the effectiveness of the applied interventions, in addition to self-reports, objective laboratory parameters are necessary to be followed. Gamma-glutamyl-transferase (GGT) and mean cellular volume (MCV) are known to be traditional and still the most effective and widely available markers for monitoring patients’ drinking habits [24]. Although MCV has low sensitivity (40%), it can be a useful marker for screening for alcohol consumption, especially when used in combination with GGT [25].

With this study, we confirm that in-hospital BI reduces alcohol consumption after an alcohol-induced AP episode.

## 2. Materials and Methods

### 2.1. Setting and Study Design

This is a post-hoc analysis of a prospective electronic data registry. In 2016, three major centers of the Hungarian Pancreas Study Group (HPSG) started to integrate BIs into hospital care for patients with alcohol-induced AP. Patients were enrolled between 2016 and 2021. The list of centers can be seen in Appendix A.

### 2.2. Patients

Altogether, 313 constitutively enrolled patients with alcohol-induced AP were checked for eligibility. Based on inclusion and exclusion criteria, 99 patients were eligible for analysis.

#### 2.2.1. Inclusion Criteria

AP was defined based on the modified Atlanta classification’s “two out of three” criteria: abdominal pain, pancreatic enzyme elevation at least three times above the upper limit, and morphological changes [26]. Alcohol-induced pancreatitis was defined as AP caused by either regular alcohol intake or consumption of an excessive amount of alcohol on one occasion. Patients who denied alcohol intake but with clear evidence of heavy drinking in their medical history and no other etiology identified were also included in this patient population.

#### 2.2.2. Exclusion Criteria

We excluded patients with alcoholic and biliary mixed etiology from the study, since biliary stones often influence GGT levels and also might cause recurrent AP independently of alcohol intake. Patients who did not present at the 1-month visit or whose admission and discharge values were not available were excluded from the analysis.

### 2.3. Intervention

Patients with alcohol-induced AP received a BI including patient education from their attending physician at least once during the patient care. The structure of the oral education includes components of BI based on the FRAMES model and a focused and goal-directed approach as in a motivational interview model, emphasizing the patients’ responsibility for their health [27]. After the assessment of the root causes of alcohol consumption, e.g., anxiety, bad thoughts, and/or boredom, the communication strategy was adapted accordingly. The interventions took 20–30 min. Additionally, leaflets were available to provide information in relation to excessive alcohol consumption, its impact on health, and options for professional help.

### 2.4. Investigated Parameters

Data on age, gender, etiology, severity, alcohol consumption (amount and frequency), previous RAP, presence of chronic pancreatitis, and in-hospital mortality were collected. Study-specific parameters GGT and MCV values were measured on admission (first 24 h), at discharge, and 1-month (23–37 days) control visit. The number of recurrent acute pancreatitis (RAP) episodes was recorded between discharge and the 1-month control. For those who were readmitted within 1 month, the readmission GGT and MCV values were analyzed. Patient questionnaires were applied to admission and at the 1-month control visit to measure alcohol consumption.

### 2.5. Outcome Parameters

The main outcome parameter was alcohol abstinence confirmed by (a) laboratory parameters (GGT and MCV levels) and (b) self-reported alcohol consumption.

### 2.6. Analysis

#### 2.6.1. Subgroups

For data analysis the cohort (*n* = 99) was divided into 2 subgroups:

Elevated GGT group (E): Patients admitted with elevated GGT levels (>50 U/L). Non-elevated GGT group (N): Patients admitted with non-elevated GGT levels.

#### 2.6.2. Statistical Analysis

Statistical analyses were performed using R 4.1 software (R Core Team; 2020.) For descriptive statistics, mean, standard deviation (SD), median, and IQR values were calculated for continuous variables, and the Wilcoxon–Mann–Whitney U test or the Kruskal–Wallis rank sum test were conducted, as applicable. For categorical variables, the Chi-square test and Fisher’s exact test were performed. For further analysis, Dunn’s post-hoc test was conducted with Benjamini–Hochberg correction and Spearman correlation was made to measure the link between the two variables. The level of significance was considered *p* ≤ 0.05.

### 2.7. Ethical Approval

The study was approved by the Scientific and Research Ethics Committee of the Hungarian Medical Research Council (17787–8/2020/EÜIG) and conducted in accordance with the Helsinki Declaration. Informed consent was obtained from all subjects involved in the study.

## 3. Results

### 3.1. Basic Characteristics and Data Quality

Altogether, 99 alcohol-induced AP cases were included in our analysis. Out of the 99 patients, 79 belonged to the elevated GGT group, while 20 cases were included in the non-elevated GGT group. Overall, 89% of the patients were male, and the mean age at admission was 50.05 ± 11.37 years. Regarding severity, 62% of the AP episodes were mild, and the median length of hospital stay was 6 (5–9) days. Details are shown in Table 1. Information on the quality of data is provided in Appendix A. The investigated cohort represented the total cohort of patients with alcohol-induced AP. Representativity analysis can be seen in Appendix A.

### 3.2. Acute Pancreatitis Is Often Followed by Another Episode

Overall, 40% of the patients had a previous AP episode. From the analyzed cohort, 14% of the patients had the diagnosis of chronic pancreatitis at admission, and in 17% of cases, hypertriglyceridemia was noted in the medical history.

### 3.3. Frequent Alcohol Drinkers Have Higher GGT Level

More than half of the admitted patients (54%) reported daily alcohol consumption, and the average amount of consumed alcohol was 81.06 ± 65.26 g. There was a significant difference in the admission GGT values between the occasional and daily drinker groups (210 ± 268 U/L vs. 267 ± 470 U/L, respectively, *p* = 0.01). More than half of the patients (66%) in the E subgroup reported daily intake, while only 5 patients (25%) reported daily consumption in the N subgroup (*p* = 0.004). There was no significant difference between admission MCV levels based on the alcohol consumption frequency. No correlation was found between alcohol consumption amount and on-admission GGT (*p* = 0.14) and MCV (*p* = 0.23). Further details are shown in Table 1.

### 3.4. Significant Decrease Is Detected in GGT Value 1-Month Following In-Hospital Patient Education

The mean value of discharge GGT in Group E was 294.00 ± 250.75 U/L, while at the 1-month control visit, 103.20 ± 113.50 U/L was measured, meaning an average decrease of 49.25 ± 74.50% (*p* < 0.001) (Appendix A, Figure 1 and Figure 2). In Group N, out of 20, only 2 patients’ GGT levels increased above the normal level at the 1-month control visit. Comparing patients based on sex, slightly more male patients reported avoiding alcohol entirely (80% vs. 72%); however, the GGT level decrease was slightly more relevant in women (decreased by 70% vs. 65%) The effectiveness of BI on serum GGT levels in patients with elevated or non-elevated admission GGT levels is visualized in Figure 1.

### 3.5. MCV Value Showed Significant Reduction 1-Month Following In-Hospital Patient Education

In Group E the mean value of MCV was 93.73 ± 5.30 fL at discharge and 92.07 ± 5.10 fL at the control visit, which means an average 1.85 ± 3.12% decrease (*p* < 0.001) (Table 1, Figure 1 and Figure 2). No one had macrocytosis (MCV > 95 fL at the control visit).

### 3.6. 75–80% of the Patients Kept Abstinence 1-Month Following In-Hospital Patient Education

Collecting self-reported alcohol intake at the control visit, most of the patients, 63/79, (80%) in Group E and 15/20 (75%) in Subgroup N, kept abstinence from alcohol. Out of the 99 analyzed patients, 3 (all belonging to Group E) were readmitted due to alcohol-induced RAP.

## 4. Discussion

Excessive alcohol consumption has a negative effect on almost every organ, causing a variety of disorders. It is generally known that alcohol abuse can lead to liver cirrhosis, but it is less common worldwide that alcoholic pancreatitis is one of the most painful and serious consequences of alcohol abuse and can subsequently lead to CP [5].

Studies investigating RAP episodes have shown that the prevalence of RAP ranges from 25 to 45% (the follow-up periods were highly different), and 80% of recurrences happen within 4 years [4,28]. In our cohort, 40% of the admitted patients had a previous AP episode, and 14% had the diagnosis of chronic pancreatitis on admission, which is consistent with a Dutch cross-sectional analysis [29]. Three patients were readmitted due to recurrence, all of whom were diagnosed with CP.

The aim of the preventive approach is to reach abstinence in this population, since the biggest risk factor of having RAP is the continuous alcohol intake in a dose-dependent manner [21]. Based on a recent cohort analysis, not only the entire abstinence but also a lower amount of alcohol consumption is related to better outcomes [12]. Pelli et al. found that none of the patients had RAP during the 24-month period and stayed totally abstinent [30]. Our results showed that among patients who reported abstinence (75%), there was no readmission due to RAP within 1 month, and the majority of these patients (95%) had decreased GGT levels on the control visit compared to the discharge value.

However, abstinence cannot be achieved as simply as it may seem at first. Therefore, particular efforts are needed to be brought into patient care in order to diminish regular alcohol consumption. Although several studies have shown the beneficial effects of BIs on alcoholic patients, their structure and frequency are still under debate [21,22,23]. Nordback et al. found better results with regular interventions at 6-month intervals in outpatient care compared to single care during hospitalization. A meta-analysis by Platt et al. showed that brief advice provided by nurses brings the most favorable outcomes, and Kaner et al. also confirmed that a longer duration of counseling probably has no relevant additional effect [22,23]. There are several undergoing randomized clinical trials investigating the effectiveness of the psychological approach on alcohol or smoking habits [11].

### 4.1. Strengths and Limitations

The strengths of our study are the multicentric uniform data collection and high data quality. The limitations are the retrospective nature of the analysis, the short follow-up time, the absence of the control group, and that we also have to consider the possible differences between BI methods since psychological interventions were performed by physicians in different centers.

### 4.2. Implications for Patients

Incorporating psychological interventions, such as BI in regular in- and outpatient care could promote abstinence and prevent recurrent episodes. Additionally, these communication methods need to be extended among practitioners, especially in the field of gastroenterology and general practice.

### 4.3. Implications for Research

Longitudinal studies and RCTs are needed to identify the adequate structure and frequency of BIs to achieve alcohol abstinence and minimize the risk of alcohol-induced RAP [11].

## 5. Conclusions

BI is an effective tool to reduce alcohol consumption and to prevent RAP. In accordance with previous observations, decreasing serum GGT values correlated with the self-reported alcohol avoidance; thus, serum GGT can be a reliable, easy-to-use clinical marker to follow patients’ drinking habits after an alcohol-induced AP episode [25,31].

## Figures and Tables

**Figure 1 nutrients-14-02131-f001:**
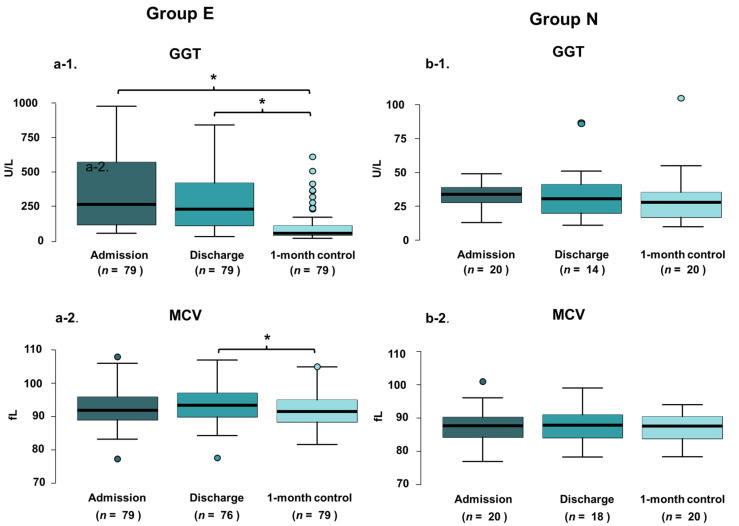
GGT and MCV values of patients in Groups E and N in different time points visualized by boxplots. E-patients with elevated on-admission GGT level (**a**); N-patients with non-elevated on-admission GGT level (**b**); GGT—gamma-glutamyltransferase; MCV—mean corpuscular volume. * *p* < 0.001.

**Figure 2 nutrients-14-02131-f002:**
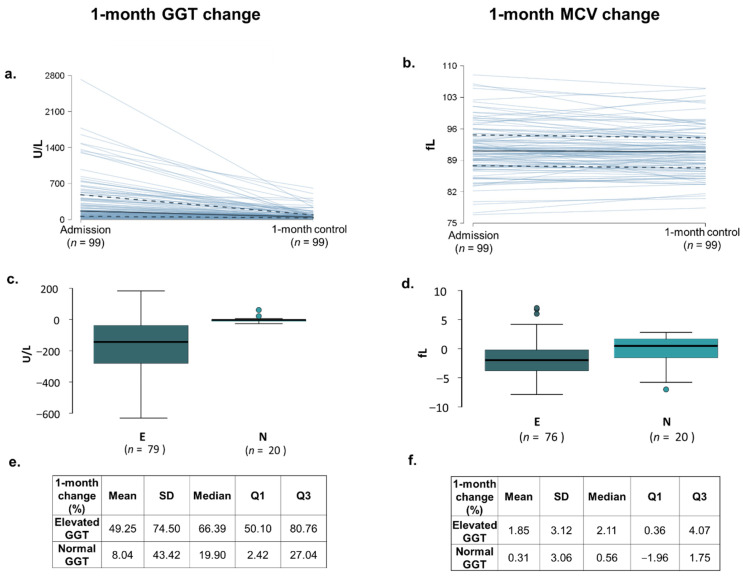
Analysis of the change in GGT and MCV levels. Figures show the change in laboratory values between discharge and the 1-month control visit. (**a**,**b**) Line chart; median ——— Q1; 3; (**c**,**d**) change in absolute value; (**e**,**f**) change in percent value. Note: in the group of patients with elevated (E) admission GGT level, discharge and 1-month values, and admission and 1-month values in the group of patients with non-elevated (N) admission GGT level, were included in the analysis. fL—femtoliter; U/L—unit/liter; E: patients with elevated admission GGT; N: patients with non-elevated admission GGT.

**Table 1 nutrients-14-02131-t001:** Summary of patient characteristics and laboratory values.

	Unit	Overall	Elevated Admission GGT	Non-Elevated Admission GGT
Patients	*n*	99	79	20
**Epidemiology**				
Gender				
Male	*n* (%)	88 (89)	68 (86)	20 (100)
Female	*n* (%)	11 (11)	11 (14)	0
Age (year)	Mean ± SD	50.05 ± 11.37	48.84 ± 11.21	54.85 ± 10.96
	Median (IQR)	50 (44–57)	50 (41–56)	54 (49–59)
**Outcomes**				
Length of hospitalization (days)	Mean ± SD	9.94 ± 10.53	9.56 ± 9.65	11.45 ± 13.66
	Median (IQR)	6 (5–9)	6 (5–9)	7 (4–10)
Severity				
Mild	*n* (%)	61 (62)	50 (63)	11 (55)
Moderate	*n* (%)	28 (28)	21 (27)	7 (35)
Severe	*n* (%)	8 (8)	6 (8)	2 (10)
**Medical history**				
Previous acute pancreatitis	*n* (%)	40 (40)	31 (39)	9 (45)
Chronic pancreatitis	*n* (%)	14 (14)	9 (11)	5 (25)
Hypertriglyceridemia	*n* (%)	17 (17)	15 (20)	2 (10)
Alcohol consumption (frequency)				
None	*n* (%)	6 (6)	5 (6)	1 (5)
Occasionally	*n* (%)	24 (24)	14 (18)	10 (50)
Monthly	*n* (%)	2 (2)	2 (3)	0 (0)
Weekly	*n* (%)	13 (13)	9 (11)	4 (20)
Daily	*n* (%)	54 (54)	49 (62)	5 (25)
Alcohol consumption (gram/occasion)	Mean ± SD	81.06 ± 65.26	84.43 ± 69.25	67.55 ± 44.88
**Laboratory parameters**				
Admission GGT (U/L)	Mean ± SD	364.57 ± 471.14	448.58 ± 493.43	32.7 ± 10.34
	Median (IQR)	166 (64–493)	263 (115.5–571)	34 (28.5–39)
Discharge GGT (U/L)	Mean ± SD	255.28 ± 248.88	294 ± 250.75	36.79 ± 23.78
	Median (IQR)	194 (70–399)	229 (108–419)	30.50 (21.25–40.25)
1-month GGT (U/L)	Mean ± SD	88.55 ± 105.80	103.2 ± 113.5	30.65 ± 20.90
	Median (IQR)	91 (87.3–94)	53 (41–108)	28 (18–35.25)
Admission MCV (fL)	Mean ± SD	91.45 ± 6.04	92.51 ± 5.69	97.29 ± 5.70
	Median (IQR)	166 (87.8–94.65)	92 (89–95.90)	35 (28.5–39)
Discharge MCV (fL)	Mean ± SD	92.58 ± 5.79	93.73 ± 5.3	97.74 ± 5.34
	Median (IQR)	92.5 (88.93–96.25)	93.4 (89.95–97.03)	87.85 (84.25–90.95)
1-month MCV (fL)	Mean ± SD	91.02 ± 5.34	92.07 ± 5.10	86.9 ± 4.22
	Median (IQR)	90.9 (87.3–94)	91.5 (88.35–95)	87.6 (83.92–90.33)
1-month GGT change (U/L)	Mean ± SD	152.67 ± 195.94	190 ± 202.16	2.05 ± 17.69
1-month GGT change (U/L; %)	Mean ± SD	40.92 ± 71.13	49.25 ± 74.50	8.04 ± 43.42
1-month MCV change (fL)	Mean ± SD	1.50 ± 2.95	1.79 ± 2.93	0.38 ± 2.83
1-month MCV change (fL; %)	Mean ± SD	1.53 ± 3.16	1.85 ± 3.12	0.31 ± 3.06
**Self-reporting**				
1-month abstinence	*n* (%)	74 (79)	60 (80)	14 (70)

GGT—gamma-glutamyltransferase; MCV—mean corpuscular volume.

## Data Availability

Access to sequencing data is available on request.

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
