# Peer review of "In-Hospital Patient Education Markedly Reduces Alcohol Consumption after Alcohol-Induced Acute Pancreatitis"

_nutrients, 2022, doi:10.3390/nu14102131_

Round 1

Reviewer 1 Report

This study exhibits that in-hospital brief intervention reduces alcohol consumption after an alcohol-induced acute pancreatitis episode. This will be a possible psychological therapy to reduce alcohol consumption and to prevent recurrent AP. As mentioned in this manuscript the short follow-up time, the absence of the control group should be the big issues here.

  1. In this study, will it be better if more female patients are selected? I am wondering whether gender make a big difference in this therapy?
  2. In 2. Patients you could just combine “Inclusion criteria” part and “Exclusion criteria” part as one part.
  3. There might be some usage issues about format and punctuation.
  4. In Figure 1. it needs to clarify the group linked to each image. Or use title like “a. b. c.” as you did in Figure 2.
  5. For the strengths and limitations of this study and some useful implications. Will it be possible to collect more feedbacks/information from these patients to support the conclusion?

Reviewer 2 Report

Substance use disorders are a major public health problem facing many countries. The ethanol is the most common substance of abuse. Excessive alcohol use can damage all organ systems, but it particularly affects the brain, heart, liver, pancreas and immune system. Women are generally more sensitive than men to the harmful effects of alcohol, primarily due to their smaller body weight, lower capacity to metabolize alcohol, and higher proportion of body fat. Therefore, in my opinion a limited numer of data presented for women is insufficient. Furthermore, the manuscript does not provide sufficient information on in-hospital pateint education. there is a need for more information on these methods.the statement „abstinence from alcohol is the best prevention against recurred episodes” included in discussion is true, but it is widely known and insufficient. Needs to present the Authors' broader observations on how to obtain abstinence.
